# A Markov Chain Approach to Multicriteria Decision Analysis with an Application to Offshore Decommissioning

Fernanda F. Moraes [1], Virgílio José M. Ferreira Filho [1], Carlos Eduardo Durange de C. Infante [1,2], Luan Santos [1,3] and Edilson F. Arruda [1,4,*]

1 Alberto Luiz Coimbra Institute for Graduate School and Research in Engineering, Federal University of Rio de Janeiro, Rio de Janeiro 21941-598, Brazil
2 Department of Business and Accounting Sciences, Universidade Federal de São João del Rei, São João del Rei 36307-352, Brazil
3 Faculty of Business and Accounting Sciences, Federal University of Rio de Janeiro, Rio de Janeiro 22290-240, Brazil
4 Department of Decision Analytics and Risk, Southampton Business School, University of Sohtampton, 12 University Rd., Highfield, Southampton SO17 1BJ, UK
* Correspondence: efarruda@pep.ufrj.br

**Abstract:** This paper proposes a novel approach that makes use of continuous-time Markov chains and regret functions to find an appropriate compromise in the context of multicriteria decision analysis (MCDA). This method was an innovation in the relationship between uncertainty and decision parameters, and it allows for a much more robust sensitivity analysis. The proposed approach avoids the drawbacks of arbitrary user-defined and method-specific parameters by defining transition rates that depend only upon the performances of the alternatives. This results in a flexible and easy-to-use tool that is completely transparent, reproducible, and easy to interpret. Furthermore, because it is based on Markov chains, the model allows for a seamless and innovative treatment of uncertainty. We apply the approach to an oil and gas decommissioning problem, which seeks a responsible manner in which to dismantle and deactivate production facilities. The experiments, which make use of published data on the decommissioning of the field of Brent, account for 12 criteria and illustrate the application of the proposed approach.

**Keywords:** multicriteria analysis; Markov chains; decommissioning

## 1. Introduction

Multicriteria decision analysis (MCDA) emerged in the 1960s to support decision making whilst accounting for distinct, possibly conflicting criteria [1]. Since then, a vast and diverse literature has presented a wide variety of methods and models to help decision makers reach a decision when no single best solution exists with respect to all criteria (e.g., [2–6]).

Some of the most commonly used MCDA models are the Analytic Hierarchy Processes (AHP) by Saaty [7], Preference Ranking Organization Method for Enrichment Evaluations (PROMETHEE) by Brans and Vincke [8], Elimination and Choice Expressing Reality (ELECTRE) by Roy [9], Multi-Attribute Utility Theory (MAUT) by Edwards and Newman [10], Simple Additive Weighting (SAW) by Fishburn [11], Additive Aggregation with Variable Interdependent Parameters (VIP analysis) by Dias and Clímaco [12], and Technique for Order Preference by Similarity to Ideal Solution (TOPSIS) by Hwang and Yoon [13]. For a comprehensive overview of these methods and their characteristics, the reader is referred to Figueira et al. [14].

Upon considering performance measures for multiple criteria, MCDA methods often prescribe the use of weights to aggregate these measures and thereby produce a ranking of the available alternatives [15]. The attribution of weights, however, is by no means an easy

task. While it carries an intuitive notion of the relative importance of each criterion, such an importance may be very difficult to quantify and highly subjective [16]. Indeed, it is often the case that the opinions and preferences of decision makers with respect to criteria are rather divergent, and averaging them may produce weights that do not capture the variety of opinions (e.g., [17–19]). Another important issue is the time required to form a consensus among stakeholders over the weights. Such a time may be excessive depending on the overall duration of the decision-making process [20] and on the number of choices to be undertaken in sequence [21]. Finally, different sets of weights can potentially generate rather distinct outcomes. Hence, the process may be rather sensitive to largely arbitrary parameters, whose evaluation may change over time or depending on the stakeholders selected to perform it [16].

The reproducibility and reliability of multicriteria analysis can also be compromised by the excess of model-specific parameters. On the one hand, these parameters are generally introduced to model personal preferences of the decision maker, especially to facilitate the comparison of similar performances with respect to a given objective; an example is the choice of a preference function in PROMETHEE. On the other hand, they compromise reproducibility and introduce noise into the process, potentially undermining the analysis and interpretability of the results [21–23]. Furthermore, the need to adjust many parameters on a case-by-case basis becomes cumbersome when the process involves a possibly large batch of decisions to be taken in a reduced time window.

The MCDA literature acknowledges the need for tools to help evaluate the stochastic effect of decision makers' preferences and improve transparency [24]. However, so far, these tools are generally limited to visualization aids and sensitivity analysis routines with respect to weight variation [25–27]. For example, Tervonen and Lahdelma [28] conducted the latter by means of a Monte Carlo simulation. Nonetheless, perhaps because the methods were originally tailored for deterministic problems, the effective treatment of uncertainties in the evaluation of the objective functions so far has received little attention in the specialized literature.

We argue that introducing uncertainty into existing deterministic models may not be the best course of action, especially considering that the methods were not originally developed to address uncertainty. Instead, we believe that by focusing on the intrinsic uncertainty of the performance measures, we can make use of probabilistic systems' theory and stochastic processes to derive an adequate MCDA framework under uncertainty. To that end, this paper models the decision-making process as a continuous-time Markov chain whose evolution is modulated by the objective functions and seamlessly accounts for uncertainty. An example for this argument is the use of MCDA methods capable of probabilistically calculating the use of uncertainty in direct relationships between alternatives. Some methods cited in the literature review better explain this point. An important point argued here is that the Markov chain makes possible the appropriate design for the continuous use of probability with uncertainty in decisions. While deterministic MCDA methods often use user-defined parameters to simulate some level of variation, the proposed method is devised for stochastic dynamics and, hence, provides a tool for decisions under uncertainty that requires no additional calibration. For more details on Markov chains and their many real-world applications, see [29,30].

In the absence of a single better solution with respect to all criteria, the decision maker will certainly experience some level of regret with the final choice. Therefore, a preemptive analysis of regrets for each possible alternative can help mitigate post-decision dissatisfaction across all criteria. Bearing that in mind, this article proposes a mini-max approach that selects the alternative that minimizes the maximum regret within the set of criteria. This approach has the advantage of not requiring weights to reconcile different criteria. Therefore, it circumvents the limitations and biases discussed earlier. It is also worth emphasizing that the proposed model is stochastic. Therefore, whilst also applicable to deterministic problems, the approach is designed to model uncertainties.

While we can discuss the strengths and drawbacks of specific models, it is clear that MCDA is an invaluable tool for supporting decision making. It provides a framework for understanding the multiple dimensions of the problem, structuring it, and incorporating the preferences of the decision maker. In this paper, we propose the combined use of the MCDA framework with Markov chains for a seamless treatment of uncertainty in the performance functions. Indeed, uncertainty is embedded in the innovative use of Markov chains, which considers decision making as a dynamic process, whereby the preference evolves in time before stabilizing. Furthermore, the steady-state probability of each alternative for a given criterion can be readily interpreted as its degree of preference for that criterion, without the need for any user-defined parameters. This facilitates reproducibility and introduces robustness by preventing parameter-selection biases.

This paper is motivated by oil and gas decommissioning problems [31], which require swift decommissioning decisions with respect to many distinct pieces of equipment disposed of in distinct environmental conditions. In such problems, time is scarce, and it is not possible to infer the preference of the decision maker with respect to each each individual piece of equipment. Scarce time also precludes the use of streamlined procedures with multiple user-defined parameters that would need prior calibration and posterior verification for each piece of equipment. To facilitate a streamlined and reproducible analysis, we propose an approach that minimizes the maximum regret over the set of criteria. For each criterion, the regret is defined as a function of the steady-state probabilities of the Markov chain and can be calculated without the need for user-defined parameters. This results in a Markov chain and regret MCDA approach that seamlessly tackles uncertainty and is free of user-defined parameters. Such an approach is particularly useful for the class of MCDA problems with uncertain performance functions and multiple decisions taken in sequence. We validate the approach in the light of a decommissioning case study. The results highlight the simplicity of the proposed model, and useful interpretations of the results are provided.

Consequently, this complex analysis can result in decisions with minimal flaws, which justifies the use of regret in decision modeling. Minimizing maximum regret turns the decision into a more robust and fairer process. This process proposes an approach using Minimax regret (minimax deviation) to address the problem of forming a neural-based classifier when, a priori, the class probabilities are partially or completely unknown, even by the end user. In addition to other adaptive approaches that address this problem, it is important to highlight those that deal with problems of uncertainty a priori by the principle of minimizing the maximum possible risk.

The remainder of this paper is organized as follows. Section 2 features a brief review of multicriteria decision analysis. Section 3 introduces the proposed Markov chain and regret (MC&R) framework and explains the modeling choices. Section 4 introduces the case study and reports the obtained results, and, finally, Section 5 concludes the paper.

## 2. Literature Review

### 2.1. Multicriteria Decision Analysis

As previously mentioned, the rationale of multicriteria decision analysis (MCDA) is to make informed decisions in the presence of multiple, possibly conflicting criteria [32]. The presence of multiple criteria gives rise to a partial ordering that often does not allow a single action to be declared optimal [33]. To circumvent this issue, MCDA techniques comprise two families of approaches designed to model preference, namely, utility-based models and the outranking methods [34,35].

Utility-based models hinge on the aggregation of different points of view into a single scalar function that must be subsequently optimized [34]. Typically, they produce a single score for each alternative by adding weighted single-criterion evaluations. Because the performance is aggregated, poor performances in one criterion can be compensated by good performance in the remaining criteria. AHP, MAUT, VIP analysis, TOPSIS, and single additive average (SAW) are examples of utility-based methods.

Introduced by [7], AHP is a purely deterministic method that organizes the criteria according to a hierarchy. For each criterion, pairwise comparisons among alternatives result in an implicit normalized ranking. The rankings are then aggregated according to the hierarchical structure and the assigned weights to produce a unified ranking. MAUT was derived from Utility Theory and Decision Theory [36]. This method models probabilistic consequences and produces a Bayesian model of expert evaluations [37]. However, the model's construction is quite involved and requires expert knowledge on the part of the analyst and specialized input from the decision maker [38]. TOPSIS defines a theoretical ideal solution that combines the best performances with respect to all criteria and uses linear algebra to find a real solution that is close enough to this ideal solution [39]. The method is deterministic and is often combined with posterior sensitivity analysis. Dubbed single additive weighting, SAW is arguably the simplest multi-criteria method. In fact, the method aggregates the criteria by means of a weighted linear sum that effectively maps the multicriteria problem into a simple monocriterion problem. The tool is rather simple and allows no treatment of uncertainty.

Proposed by Dias and Clímaco [12], VIP analysis is based on the progressive reduction of the number of alternatives by the elimination of those considered dominated. This method draws conclusions by solving linear programs that determine the worst and the best possible values for each action. An advantage of this approach is the fact that it does not require precise values, accepting linear intervals and restrictions as inputs, which can be helpful to the decision maker when in doubt. Despite the greater flexibility for the decision maker, there is still a dependency on arbitrary parameters, as they must quantify their preferences.

The outranking methods, such as ELECTRE and PROMETHEE, are called non-compensatory because they strive to mitigate compensation by introducing preference and indifference relationships based on arbitrary parameters. They rely on pairwise comparisons between alternatives for each criterion, which either reinforce or contradict the superiority of an alternative with respect to another and are weighted according to the criterion under consideration.

The ELECTRE family includes a series of methods that make use of two basic concepts, namely, agreement and disagreement. The agreement indices reinforce the superior performance of an alternative, whereas disagreement indicates the opposite. The weighted average of these indices gives rise to an overall ranking of alternatives. The method was devised for deterministic problems.

PROMETHEE is a family of methods that also make use of pairwise comparisons and assign weights to criteria [40]. In order to consider similar performances, the method makes use of preference functions, which are used to arbitrarily define indifference levels. While this is an interesting characteristic for a single decision, it is largely arbitrary and compromises reproducibility in large-scale processes, where parameters have to be tuned for each individual run. Moreover, the effect of changes in these parameters is difficult to estimate and can undermine the reliability [21,22]. Despite the similarity with ELECTRE methods, a significant difference is that PROMETHEE does not use the concept of disagreement.

### 2.2. Uncertainty Analysis

Multicriteria analysis is subject to uncertainty (*i*) in the parameters, which tend to be subjective [41], (*ii*) in the assessment of each alternative with respect to each criterion [42], or even (*iii*) in interactions between criteria [43]. As the number of parameters varies according to the selected approach (see Section 2.1), so does the number of sources of uncertainty [44].

Concerning MCDA methods, uncertainty may stem from inaccurate information or inaccurate or biased assessments by decision makers [23,42]. It may also be due to non-deterministic and unpredictable consequences of certain available alternatives, or even due to biases in establishing the model parameters [41]. Note that the latter become more

significant as the number of parameters increases. The reader is referred to [44] for a comprehensive account of the treatment of the sources of uncertainty in MCDA models.

MAUT is based on preference modeling concepts and admits strict preference and indifference based on an aggregate utility function. Uncertainty is typically addressed by evaluating the expected reward, which is sometimes estimated with Monte Carlo methods due to the combinatorial explosion in the number of potential uncertainties. These are due not only to the stochastic response to the alternatives, but also to the subjective modeling parameters [44,45]. However, an effective evaluation demands a significant number of samples.

Stochastic Multicriteria Acceptability Analysis (SMAA) also resorts to Monte Carlo simulations to exploit the variation of the parameters of MCDA techniques. The objective, however, is to find sets of weights (parameters) for which each alternative would be preferred in the respective MCDA model [46]. The stochastic component is a Monte Carlo sampling technique applied to exploit the space of weights (parameters), given that a full exploration of this space is computationally intractable. This technique is employed to analyze the effects of weight (parameter) changes in connection with ELECTRE [6], PROMETHEE [41], and AHP [42].

Introduced by [47], Simulated Uncertainty Range Evaluations (SUREs) allow decision makers to prescribe preferred intervals for the model parameters. The method makes use of triangular distributions for all uncertain parameters and evaluations and gives rise to a kernel density plot that conveys the preferences and the overlapping uncertainties regarding the available alternatives.

Ref. [48] used Dempster–Shafer (DS) evidence theory to address the uncertainties regarding subjective evaluations of pairwise comparisons in AHP. These evaluations were considered or disregarded based on yet another user-defined parameter, a measure of plausibility arbitrated by the decision maker. Such a measure was modeled via fuzzy numbers in the works of Sadiq and Tesfamariam [49] and Durbach and Stewart [44]. Finally, Yang et al. [50] also resorted to fuzzy sets to model imprecise specialist evaluations. The distinctive feature of their work is the use of a one-iteration Delphi process to reconcile conflicting intervals.

Both interval analysis and fuzzy numbers were also applied to emulate a certain level of uncertainty in the TOPSIS method [22,51]. Whilst the former work described the weight of each criterion as a fuzzy set, the latter modeled this parameter as an interval. In another line of research, Engau and Sigler [52] strove to extend the notion of Pareto optimality to multicriteria optimization under uncertainty.

### 2.3. Probabilistic MCDA Approaches

The combination of the Markov chain with multicriteria analysis is used as a tool to improve the outcome of multicriteria analysis. Different from the proposal of the article, when used together the Markov chain calculates the performance of the alternatives while the comparison between the alternatives is made by the deterministic multicriteria method, showing improvement in the result.

Ref. [53] discussed the need for a proper stochastic characterization of uncertain expert evaluations in AHP, as these may be subject to errors, and they proposed an odds ratio approach in order to model pairwise comparisons. This resulted in a multinomial distribution of preferences that warranted probabilistic characterizations of the alternatives by means of hypothesis testing. In particular, the paper applied Monte Carlo Markov chains to find a preference distribution for the alternatives by sampling the weights from the ensuing multinomial distribution.

While employing a deterministic MCDA approach, the authors of [54,55] applied Markov chains in order to evaluate individual maintenance policies in healthcare organizations. A deterministic MCDA analysis by means of MACBETH followed, which compared the alternatives based on their steady-state performances. Similarly, the authors of [56] used a discrete-time Markov chain to evaluate the average performance of each alternative

and model the dynamic systems' behavior. The alternatives were then compared by means of a TOPSIS approach with user-defined weights.

In contrast, the authors of [57] made use of Markov chains to find user preference patterns in cloud services and employed deterministic MCDA techniques to rank the available services. The identified patterns were then used for the recommendation and selection of the suitable services. Finally, Markov chains with Gauss–Jordan elimination were applied as an accessory to speed up the solution process of a deterministic generalized ANP formulation [58]. The 'power' matrix method, a procedure necessary for the stability of the decision system, is one of the critical calculations in the mathematical part of the method. In this article, the results are similar to the power matrix method, with less time and number of calculations being required and dealing with cyclic super matrices effectively.

In this work, we propose a novel purely stochastic approach to multicriteria decision making. In contrast to [53], who introduced a statistical formulation as a post-processing technique for a deterministic MCDA approach, we model the decision process as a stochastic process. Hence, our model avoids the shortcomings of deterministic MCDA approaches, namely, the excess of user-defined parameters, which results in endogenous uncertainty and potentially compromises reliability and reproducibility. At the same time, the proposed formulation innovates by providing an adequate framework for the treatment of uncertain performance functions and scores with respect to any number of criteria.

The models described here allow a better fit in understanding the relationship between uncertainties and dimensions for complex decision making. Comparing the MCDA methods with the Markov chain proposed in this article, it is understood that the proposed model is innovative and allows, according to the literature, an advance in the treatment of sensitive data. The literature review allowed us to understand that the Markov chain has an enormous potential to satisfy the problems between uncertainties and complex parameters for decisions.

## 3. Mathematical Formulation

This section introduces the proposed Markov chain and regret (MC&R) model. Consider a multicriteria decision problem where a number of alternatives have to be evaluated with respect to a set of criteria. Let $S = \{1, 2, \ldots, m\}$ be the set of alternatives to be considered and let $C = \{1, 2, \ldots, n\}$ be the set of criteria. For each alternative $a \in S$, let $f_j(a)$ denote the performance of alternative $a$ with respect to criterion $j$. Ideally, the decision maker searches for the solution of the problem below:

$$\text{Maximise } f_j(a), \ a \in S, \ j \in C. \tag{1}$$

However, such a solution very seldom exists. Hence, the task of the decision maker is to select an action $a$ whose evaluation provides an adequate compromise among all criteria $j \in C$. To aid such a decision, we model the system by means of a series of continuous-time Markov chains $X_t^j$, $t \geq 0$, one for each criterion $j$. For a comprehensive description of continuous-time Markov chains, the reader is referred to Brémaud [29].

For each $j \in C$, the stochastic process $X_t^j$, $t \geq 0$ takes values from the set $S$ of available alternatives, evolves in continuous time, and represents the dynamic changes in the preferences of the decision maker. Observe that we model the choice as a dynamical system, in which the preference changes according to transition rates that depend upon the evaluation of all the available alternatives in the set $S$. If $X_t^j = a$, $a \in S$, at a given time $t \geq 0$, it means that $a$ is the preferred alternative with respect to criterion $j$ at time $t$. The preference changes according to a continuous-time Markov chain, and the transition rate depends upon the current state. The preference switches from alternative $a$ to alternative $b$ based on

the relative performance of these alternatives with respect to criterion $j$. The transition rate from alternative $a$ to alternative $b$ is denoted by $\lambda_j(a, b)$ and is given by Equation (2):

$$\lambda_j(a, b) = E\left(\frac{f_j(b)}{f_j(a)}\right), \ a, b \in S, \ j \in C, \tag{2}$$

where $E$ denotes the expected value. Observe that if $a$ and $b$ have similar performances with respect to criterion $j$, then $\lambda_j(a, b) \approx \lambda_j(b, a) \approx 1$, which indicates indifference. Hence, the fraction on the right-hand side of Equation (2) can be regarded as a natural and objective preference function. We stress that it does not involve arbitrary parameters or subjective evaluations by the decision makers. In addition, as a probabilistic function, it is designed to seamlessly tackle uncertainty.

Equation (2) implies an infinitesimal generator for process $X_t^j$, $t \geq 0$ given by [29]:

$$\Lambda_j = [\lambda_j](a, b) = \begin{cases} \lambda_j(a, b), & \text{if } a \neq b; \\ -\sum_{c \neq a} \lambda_j(a, c), & \text{if } a = b. \end{cases} \tag{3}$$

It follows from Equations (2) and (3) that process $X_t^j$, $t \geq 0$, will spend more time on alternatives with higher performance in terms of the criterion $j$. However, in order to calculate the exact probability that each alternative $a \in S$ will be preferred in the long-term, we need to evaluate the steady-state behavior of process $X_t^j$, $t \geq 0$. Standard Markov chain results yield that the steady-state probabilities satisfy the following system of equations [29]:

$$\begin{aligned} \pi_j \Lambda &= 0, \\ \sum_{a \in S} \pi_j(a) &= 1. \end{aligned} \tag{4}$$

The value $\pi_j(a)$ in Equation (4) can be interpreted as the probability that the decision maker will prefer alternative $a$ under criterion $j$ in the long term. Therefore, $\pi_j(a)$ is the *degree of preference* of alternative $a$ under criterion $j$. Observe that under the proposed formulation, the degree of preference is a straightforward interpretation of the steady-state behavior.

To illustrate this issue, let us suppose $S = \{a, b\}$ and assume that we have $f_j(a) = 100$ and $f_j(b) = 95$ for some $j \in C$. This yields:

$$\Lambda_j = \begin{bmatrix} -\frac{95}{10} & \frac{95}{100} \\ \frac{100}{95} & -\frac{100}{95} \end{bmatrix}.$$

The solution to (4) is $\pi_j = [0.526 \ 0.474]$. Hence, the degree of preference of alternative $a$ is $\pi_j(a) = 0.526$, whereas the degree of preference for alternative $b$ is $\pi_j(a) = 0.476$, which are quantities that can be promptly interpreted after objective calculations. In contrast, a classical non-compensatory approach such as PROMETHEE would require an arbitrary preference function, as well as arbitrary indifference and preference thresholds, to derive a similar preference quantification that would only be valid for this specific set of parameters (e.g., [59]).

In order to aggregate the degrees of preference assigned individually for each criterion $j \in C$, we propose the use of a *mini-max* approach that seeks seeks the alternative that minimizes the maximum regret. We calculate the regret with respect to the performance of the alternative with the highest degree of preference for each criterion $j$, defined as:

$$g_j^* = \max_{a \in S} \pi_j(a), \ j \in C, \tag{5}$$

where $\pi_j$ is the solution to (4) and alternative $a_j^* = \arg\max_{a \in S} \pi_j(a)$ belongs to the set of preferred alternatives under criterion $j$.

Let $R = [r_{ja}]$, $j \in C$, $a \in S$ be a matrix of regrets such that

$$r_{ja} = g_j^* - \pi_j(a), \ a \in S, \ j \in C. \tag{6}$$

The maximum regret of alternative $a \in S$ is defined as

$$r_m(a) = \max_{j \in C} r_{ja}, \tag{7}$$

and it determines the largest mismatch between the degree of preference of alternative $a$ and that of the preferred alternative over the set of performance criteria $C$.

The MC&R method then selects alternative $a^*$, which minimizes the maximum regret and, therefore, satisfies:

$$a^* = \arg \min_{a \in S} r_m(a). \tag{8}$$

## 4. Numerical Experiments

We start this section with a simple example to better illustrate the approach. Let us consider the example in Section 3, where $S = \{a, b\}$. Assume that $C = \{1, 2, 3\}$ is the set of criteria and that the solution to Equation (4) yields $\pi_1 = [0.526 \ 0.474]$, $\pi_2 = [0.643 \ 0.357]$, and $\pi_3 = [0.279 \ 0.721]$.

Then, Equation (5) yields $g_j^* = [0.526 \ 0.643 \ 0.721]$ as the vector of the highest degrees of preferences for all criteria. Now, by applying Equation (6), we obtain the matrix of regrets:

|       |           | Alternative | |
|-------|-----------|-------------|-------|
| $R =$ | **Criterion** | a       | b     |
|       | 1         | 0           | 0.052 |
|       | 2         | 0           | 0.286 |
|       | 3         | 0.442       | 0     |

Hence, the solution to Equation (7) is obtained by simple inspection of matrix $R$ above, and it yields the maximum regrets $r_m(a) = 0.442$ and $r_m(b) = 0.286$. Hence, Equation (8) yields $a^* = b$, and alternative $b$ is selected because it minimizes the maximum regret.

### 4.1. An Example with Uncertain Performance Functions

To illustrate the method under an uncertain performance function, consider an example with $S = \{a_1, a_2, a_3\}$, $C = \{1, 2\}$ and the following performance evaluations:

$$P(f_1(a_1) = w) = \begin{cases} \frac{1}{3}, & \text{if } w = 1 \\ \frac{1}{2}, & \text{if } w = 2, \\ \frac{1}{6}, & \text{if } w = 3 \end{cases} \quad f_1(a_2) = 2, \quad f_1(a_3) = 1.5,$$

$$f_2(a_1) = 1, \ f_2(a_2) = 2, \ f_2(a_3) = 3.$$

For criterion 1, a simple application of Equation (2) yields:

$$\lambda_1(a_1, a_2) = \frac{1}{3} \cdot \frac{2}{1} + \frac{1}{2} \cdot \frac{2}{2} + \frac{1}{6} \cdot \frac{2}{3} = \frac{23}{18}.$$

After applying Equation (2) to all possible pairs of actions, we obtain:

$$\Lambda_1 = \begin{bmatrix} -\dfrac{161}{72} & \dfrac{23}{18} & \dfrac{23}{24} \\[2ex] \dfrac{11}{12} & -\dfrac{5}{3} & \dfrac{3}{4} \\[2ex] \dfrac{11}{9} & \dfrac{4}{3} & -\dfrac{23}{9} \end{bmatrix}.$$

Hence, Equation (4) yields the degrees of preference of actions $a_1$ to $a_3$ with respect to criterion 1, given by $\pi_1 = [0.3147\ 0.4386\ 0.2467]$. Similarly, we have for the second criterion:

$$\Lambda_2 = \begin{bmatrix} -5 & 2 & 3 \\ 0.5 & -2 & 1.5 \\ \dfrac{1}{3} & \dfrac{1}{2} & -\dfrac{5}{6} \end{bmatrix},$$

and $\pi_2 = [0.0701\ 0.2420\ 0.6879]$. Equation (6) yields the matrix of regrets:

$$R \;=\;$$

| Criterion | Alternative | | |
|:---:|:---:|:---:|:---:|
| | $a_1$ | $a_2$ | $a_3$ |
| 1 | 0.1239 | 0 | 0.1919 |
| 2 | 0.6178 | 0.4459 | 0 |

Finally, Equation (8) implies $a^* = a_3$, since this is the alternative that minimizes the maximum regret with $r_1(a_3) = 0.1919$.

### 4.2. Application to Decommissioning of Oil and Gas Fields

For a more thorough illustration of the proposed method, we selected the decommissioning process of an oil and gas production facility in the Brent field [60], which is located in the East Shetland Basin of the North Sea, 186 km northeast of Lerwick, Shetland Islands, Scotland, and 140 m below the surface of the ocean. The program is the culmination of a 10-year study that mobilized academics and stakeholders, and it includes detailed recommendations for closing and securing the four platforms and sub-sea infrastructures.

For the experiment, we selected a specific rigid pipeline labeled *PL002/N0201* in [60]. The decision was made by considering a set *C* comprised of 12 criteria, eight of which received quantitative evaluations, whereas the remaining four received qualitative evaluations from experts. Table 1 depicts the set of criteria.

**Table 1.** Sub-criteria for the decommissioning of the Brent field.

| Area | Label | Unit | Criterion |
|:---:|:---:|:---:|:---:|
| | 1 | PLL | Safety risk to offshore project personnel |
| Safety | 2 | PLL | Safety risk to other users of the sea |
| | 3 | PLL | Safety risk to onshore project personnel |
| | 4 | Score | Operational environmental impacts |
| Environmental | 5 | Score | Legacy environmental impacts |
| | 6 | GJ | Energy use |
| | 7 | tons of $CO_2$ | Emissions |
| Technical | 8 | Score | Technical feasibility |
| | 9 | £ | Effects on commercial fisheries |
| Social | 10 | Employees Year | Employment |
| | 11 | Score | Communities |
| Economic | 12 | $10^6$ £ | Cost |

Source: Adapted from [60]. PLL stands for probability of loss of life, GJ for giga Joules, £ for pounds sterling, and $CO_2$ for carbon dioxide.

Table 2 conveys the available decommissioning alternatives. A single alternative is to be selected by considering the set of twelve criteria described in Table 1. These 12 criteria are essential and sufficient for understanding the composition of parameters in the context of sustainability in decommissioning studies.

**Table 2.** Pipeline decommissioning alternatives.

| Alternatives | Description |
|---|---|
| $a_1$ | Leave tied-in at platform; remote and trenched |
| $a_2$ | Leave tied-in at platform; remote and rock-dumped |
| $a_3$ | Disconnect from the installation; trench and backfill whole length |
| $a_4$ | Disconnect from the installation; rock-dump whole length |
| $a_5$ | Recover whole length by cutting and lifting |
| $a_6$ | Recover whole length with a reverse S-lay |

Source: Adapted from [60].

Table 3 depicts the performance of each available alternative with respect to each criterion.

**Table 3.** Performance of the alternatives with respect to each criterion.

| Criterion | Alternatives | | | | | |
|---|---|---|---|---|---|---|
| | $a_1$ | $a_2$ | $a_3$ | $a_4$ | $a_5$ | $a_6$ |
| 1 | 625.00 | 625.00 | 416.67 | 500.00 | 357.14 | 232.56 |
| 2 | 12.50 | 10.66 | 625.00 | 625.00 | 833.33 | 833.33 |
| 3 | 833.33 | 833.33 | 833.33 | 833.33 | 833.33 | 833.33 |
| 4 | 0.99 | 0.98 | 0.91 | 0.85 | 0.91 | 0.95 |
| 5 | 0.95 | 0.95 | 1 | 0.90 | 1 | 1 |
| 6 | $4.11 \times 10^{-5}$ | $4.79 \times 10^{-5}$ | $4.11 \times 10^{-5}$ | $4.11 \times 10^{-5}$ | $4.11 \times 10^{-5}$ | $3.59 \times 10^{-5}$ |
| 7 | $5.3 \times 10^{-4}$ | $5.3 \times 10^{-4}$ | $4.5 \times 10^{-4}$ | $5.3 \times 10^{-4}$ | $6.3 \times 10^{-4}$ | $5.3 \times 10^{-4}$ |
| 8 | 0.92 | 1 | 0.80 | 1 | 0.84 | 0.35 |
| 9 | 0.00 | 0.00 | 228,344.2 | 183,367.3 | 228,344.2 | 228,344.2 |
| 10 | 6.35 | 3.17 | 9.53 | 6.35 | 19.05 | 31.76 |
| 11 | 1 | 1 | 1 | 1 | 0.95 | 0.95 |
| 12 | 0.75 | 0.93 | 0.47 | 0.53 | 0.20 | 0.13 |

Source: Adapted from [60].

Now, the first step is the construction of the infinitesimal generators for all 12 criteria by making use of Equation (3). For example, for criterion 1, we have:

$$
\Lambda_1 = \begin{bmatrix}
-3.41 & 1 & 0.67 & 0.80 & 0.57 & 0.37 \\
1 & -3.41 & 0.67 & 0.80 & 0.57 & 0.37 \\
1.50 & 1.50 & -5.62 & 1.20 & 0.86 & 0.56 \\
1.25 & 1.25 & 0.83 & -4.51 & 0.71 & 0.47 \\
1.75 & 1.75 & 1.17 & 1.40 & -6.72 & 0.65 \\
2.69 & 2.69 & 1.79 & 2.15 & 1.54 & -10.85
\end{bmatrix}
$$

After solving (4) for each criterion, we obtain:

$$\pi_1 = [0.282 \quad 0.282 \quad 0.125 \quad 0.180 \quad 0.092 \quad 0.039];$$
$$\pi_2 = [0.000 \quad 0.000 \quad 0.180 \quad 0.180 \quad 0.320 \quad 0.320];$$
$$\pi_3 = [0.167 \quad 0.167 \quad 0.167 \quad 0.167 \quad 0.167 \quad 0.167];$$
$$\pi_4 = [0.188 \quad 0.184 \quad 0.159 \quad 0.138 \quad 0.159 \quad 0.173];$$
$$\pi_5 = [0.161 \quad 0.161 \quad 0.178 \quad 0.144 \quad 0.178 \quad 0.178];$$
$$\pi_6 = [0.163 \quad 0.222 \quad 0.163 \quad 0.163 \quad 0.163 \quad 0.125];$$
$$\pi_7 = [0.162 \quad 0.162 \quad 0.119 \quad 0.162 \quad 0.233 \quad 0.162];$$
$$\pi_8 = [0.196 \quad 0.232 \quad 0.148 \quad 0.232 \quad 0.164 \quad 0.028];$$
$$\pi_9 = [0.001 \quad 0.001 \quad 0.251 \quad 0.232 \quad 0.263 \quad 0.251];$$
$$\pi_{10} = [0.026 \quad 0.007 \quad 0.059 \quad 0.026 \quad 0.234 \quad 0.650];$$
$$\pi_{11} = [0.172 \quad 0.172 \quad 0.172 \quad 0.172 \quad 0.156 \quad 0.156];$$
$$\pi_{12} = [0.280 \quad 0.439 \quad 0.110 \quad 0.144 \quad 0.020 \quad 0.008].$$

The regret matrix $R$ produced by Equation (6) appears below, where each row corresponds to an individual criterion:

$$R = \begin{bmatrix} 0 & 0 & 0.157 & 0.101 & 0.190 & 0.243 \\ 0.320 & 0.320 & 0.140 & 0.140 & 0 & 0 \\ 0 & 0 & 0 & 0 & 0 & 0 \\ 0 & 0.004 & 0.029 & 0.049 & 0.029 & 0.015 \\ 0.017 & 0.017 & 0 & 0.034 & 0 & 0 \\ 0.059 & 0 & 0.059 & 0.059 & 0.059 & 0.097 \\ 0.071 & 0.071 & 0.114 & 0.071 & 0 & 0.071 \\ 0.036 & 0 & 0.083 & 0 & 0.068 & 0.203 \\ 0.261 & 0.261 & 0.012 & 0.031 & 0 & 0.012 \\ 0.624 & 0.643 & 0.591 & 0.624 & 0.416 & 0 \\ 0 & 0 & 0 & 0 & 0.017 & 0.017 \\ 0.159 & 0 & 0.329 & 0.295 & 0.419 & 0.431 \end{bmatrix}.$$

Hence, the vector of maximum regrets—Equation (7)—is

$$r_m = [0.624 \quad 0.643 \quad 0.591 \quad 0.624 \quad 0.419 \quad 0.431].$$

Hence, the alternative that minimizes the maximum regret according to Equation (8) is alternative $a_5$, namely, "Recover whole length by cutting and lifting", as described in Table 2. This alternative is considered the best alternative from the methods used and from the interactive analysis of the alternatives to the criteria, considering non-compensatory performances within the scope of decommissioning.

### 4.3. Analysis of the Result

In the original example, which allows a comparison of results, Shell employed a single additive average (SAW) method that was referred to as a comparative assessment [60]. They normalized the scores of each alternative and then produced a unified score for each alternative as the weighted average of the scores over the set of criteria. This resulted in a simple and easy-to-use method that is very reliant on the arbitrary weights attributed to the criteria.

The use of such methods allows a less robust and more compensatory decision, which is not ideal. On the one hand, comparative assessment allows data to be less interactive; on the other hand, the Markov method allows non-compensation between alternatives, mainly with the use of regret analysis.

The results of this paper illustrate the importance of selecting a robust decision method whose compromises are clear and easy to understand. As illustrated in [60], despite the careful discussion regarding the weights, the output of the method was still disputed, and another round was necessary to select the alternative. While another round may not be a significant barrier for a single decision, the process can become quite cumbersome in a large-scale problem, such as the decommissioning of a very large quantity of pieces of equipment that were previously used in oil and gas exploration, which typically requires an individual decision for each piece of equipment.

In contrast, the objective of the proposed approach, namely, to select the alternative that minimizes the maximum regret, is clear and easily understood by all stakeholders. Therefore, it offers a more balanced compromise that does not rely on arbitrary user-defined param- eters that, in turn, may lead to an inconsistent choice followed by a dispute among the stakeholders, as illustrated in the example.

## 5. Conclusions

This paper addresses important shortcomings of mainstream multicriteria decision analysis tools, namely, the excess of user-defined and model-specific parameters and the absence of an effective methodology for seamlessly dealing with uncertainties. These shortcomings lead to cumbersome analyses that are difficult to understand and replicate; therefore, they undermine the reliability and reproducibility of MCDA tools.

In order to improve the transparency of the decision-making process while also promoting an adequate and natural treatment of uncertainty, we introduce the Markov chain and regret (MC&R) method. The proposed MCDA approach uses a Markov chain model that produces a degree of optimality for each action with respect to each objective based on the steady-state probabilities. By selecting an action with a sub-optimal degree of optimality under any objective, the decision maker incurs a regret with regard to that objective. An action is deemed optimal if it minimizes the maximum regret in the set of criteria.

The main innovation is the modeling of the decision process by means of a stochastic process, which provides a natural way of treating uncertainty. Moreover, the approach is robust in that it does not rely on weights and user-defined parameters. Instead, it leads to a very natural way of evaluating the degree of preference of each action in terms of the long-term probability of selecting such an action.

To illustrate the application, we introduced a real-world example from the field of oil and gas decommissioning. The example demonstrates the simplicity and robustness of the proposed method and shows how we can produce a complete picture of the performance of each action with regard to each objective without introducing human bias and user-defined or method-specific parameters.

Thus, the article introduces an important question about MCDA methods and their correlates. Some aspects still need to be better explored, such as the adequacy of qualitative parameters, the relationships of uncertainty between attributes, and the preferences between alternatives. Even so, the detailed Markov chain enabled innovation in the multicriteria decision methodology. For future research opportunities, we recommend applying this method in other research contexts and in conjunction with other MCDA methods, such as PROMETHEE.

**Author Contributions:** Conceptualization, F.F.M. and E.F.A.; methodology, F.F.M. and E.F.A.; validation, F.F.M., E.F.A. and V.J.M.F.F.; formal analysis, F.F.M., E.F.A., V.J.M.F.F. and C.E.D.d.C.I.; investigation, F.F.M., E.F.A., V.J.M.F.F. and C.E.D.d.C.I.; data curation, F.F.M., E.F.A., V.J.M.F.F., C.E.D.d.C.I. and L.S.; writing—review and editing, F.F.M., E.F.A., V.J.M.F.F., C.E.D.d.C.I. and L.S.; supervision, E.F.A. All authors have read and agreed to the published version of the manuscript.

**Funding:** Coordenação de Aperfeiçoamento de Pessoal de Nível Superior—Brasil (CAPES)—Finance Code 001. The authors would like to thank the Carlos Chagas Filho Foundation for Research Support in Rio de Janeiro, FAPERJ, for supporting the research by means of Grant E-26/202.789/2015. This

work was partially supported by the National Council for Scientific and Technological Development—CNPq, under Grants 307126/2017-0 and 311075/2018-5.

**Institutional Review Board Statement:** Not applicable.

**Informed Consent Statement:** Not applicable.

**Data Availability Statement:** Not applicable.

**Acknowledgments:** To thank the funding received and the partnership between the universities involved.

**Conflicts of Interest:** The authors declare no conflict of interest.

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
