# Peer review of "A Markov Chain Approach to Multicriteria Decision Analysis with an Application to Offshore Decommissioning"

_sustainability, doi:10.3390/su141912019_

Round 1

Reviewer 1 Report

In this paper, the authors proposed a novel approach that makes use of continuous time Markov chains and regret functions to find an appropriate compromise in the context of multicriteria decision analysis (MCDA). The proposed approach avoids the drawbacks of arbitrary user-defined and method-specific parameters by defining transition rates that depend only upon the performances of the alternatives.

After a brief review of the related literatures, I find that the current work is an original and innovative work. The motivation of the paper is obvious, and the proposed technique is correct. The work can help solve some theoretic and practice problems. Thus I suggest an acceptance. 

Author Response

Thank you. 

Reviewer 2 Report

The paper proposes a novel approach that makes use of continuous-time Markov chains and regret functions to find an appropriate compromise in the context of multicriteria decision analysis (MCDA). The proposed approach avoids the drawbacks of arbitrary user-defined and method-specific parameters by defining transition rates that depend only upon the performances of the alternatives.

The paper is very well motivated and the proposed method is critical to advancing the theory and practice of multicriteria decision analysis. My recommendation is to accept this paper after minor revisions. 

(1) Regarding your statement, "We argue that introducing uncertainty into existing deterministic models may not be the best course of action, especially considering that the methods were not originally developed to address uncertainty." I would suggest to provide some simple examples to demonstare your arguemnts. 

(2) I would suggest providing a table to analyze the literature review. What are the research gaps? How can you fill them? Compare the existing models in the literature and why you think your model is merit in the literature? Please work on improving the clarity of your paper. 

(3) There are several grammatical mistakes and typos. Please address them carefully. 

Author Response

Thank you. 

Reviewer 3 Report

1. I consider that the topic is and scientifically interesting and the manuscript is easy to follow and the terminology is appropriate to the subject.

2. The content of the paper is succinctly described and contextualized in relation to the presented theoretical background.

3. I would recommend to emphasize more in the abstract the relevance, originality and quality of the research, persuasively suggesting to the potential reader the items of interest that the work proposes.

4. I recommend the authors to present in a much clearer manner and articulate the context in which the problem is formulated in the introductory part of the paper.

5. A more rigorous methodology section should be included. Section 3 of the paper, Mathematical formulation  should be improved (pages 9-11). I recommend that you present the research method much more clear and in detail, providing the necessary elements for the reproduction of research by any other research group that uses it exactly (the repetitive and reproducible nature of science).

6. In the case of section 4 Numerical experiments, justify the choice of the 12 criteria presented in table 1, page 14. Are the 12 criteria sufficient for the analysis of the proposed research objective? Are these criteria relevant?

7. It is known that in the case of decommissioning of oil land gas fields the number of equipment is very large. In practice, decommissioning operations influence each other. To what extent is the proposed method effective and appropriate for the decision-making process?

8. The concluding elements of the paper are represented by strong statements based on scientific arguments that are presented clearly and concisely. However, I believe that the authors should reflect the extent to which the results answered the questions mentioned in the introductory part: What is the research gab and what is your paper's contribution/ innovation for the research? In my opinion, solid arguments on the conclusions of the paper will open new research directions and lead to the deepening of the issues studied by potential readers.

9. We found that bibliographic references (in total of 62) are described accurately, honestly and deontologically by the authors.

10. I recommend the authors to present in a more promising manner the future research opportunities which are considered to be feasible and scientifically fertile in the field.

Author Response

Thank you.
